# ENCODING WORD ORDER IN COMPLEX EMBEDDINGS

**Benyou Wang** [*]
University of Padua
wang@dei.unipd.it

**Donghao Zhao** [*]
Tianjin University
zhaodh@tju.edu.cn

**Christina Lioma**
University of Copenhagen
chrh@di.ku.dk

**Qiuchi Li**
University of Padua
qiuchili@dei.unipd.it

**Peng Zhang** [†]
Tianjin University
pzhang@tju.edu.cn

**Jakob Grue Simonsen**
University of Copenhagen
simonsen@di.ku.dk

## ABSTRACT

Sequential word order is important when processing text. Currently, neural networks (NNs) address this by modeling word position using position embeddings. The problem is that position embeddings capture the position of individual words, but not the ordered relationship (e.g., adjacency or precedence) between individual word positions. We present a novel and principled solution for modeling both the global absolute positions of words and their order relationships. Our solution generalizes word embeddings, previously defined as independent vectors, to continuous word functions over a variable (position). The benefit of continuous functions over variable positions is that word representations shift smoothly with increasing positions. Hence, word representations in different positions can correlate with each other in a continuous function. The general solution of these functions is extended to complex-valued domain due to richer representations. We extend CNN, RNN and Transformer NNs to complex-valued versions to incorporate our complex embedding (we make all code available). Experiments [1] on text classification, machine translation and language modeling show gains over both classical word embeddings and position-enriched word embeddings. To our knowledge, this is the first work in NLP to link imaginary numbers in complex-valued representations to concrete meanings (i.e., word order).

## 1 INTRODUCTION

When processing text, the sequential structure of language is important, but can be computationally costly to model with neural networks (NNs) (Socher et al., 2011) due to the difficulty in parallelization. This has been alleviated by modeling word sequence not on the NN architecture level, but by adding *position embeddings* on the feature level. This has been done by the convolutional sequence model (ConvSeq) (Gehring et al., 2017) and the Transformer model (Vaswani et al., 2017) that replaces recurrent and convolution operations with purely attention mechanisms. More generally, vanilla position embeddings (Gehring et al., 2017) assume that individual word positions are independent and do not consider relations between neighbouring word positions. We posit that *both* the global absolute positions of words *and* their inner sequential and adjacent relationships are crucial in language. This is supported by recent empirical findings by Shaw et al. (2018) and Dai et al. (2019) who show the importance of modeling distance between sequential elements, and explicitly use extra relative position encodings to capture the relative-distance relationship of words.

We present a novel and principled approach to model both the global absolute positions of words and their inner sequential and adjacent relationships as follows: we extend each word embedding, previously defined as an independent vector, as a continuous function over an independent variable i.e., position. The benefit of continuous functions over variable positions is that word representations shift smoothly with increasing positions. Hence, word representations in different positions

---

[*] First two authors contributed equally.

[†] Corresponding author: pzhang@tju.edu.cn

[1] The code is on https://github.com/iclr-complex-order/complex-order

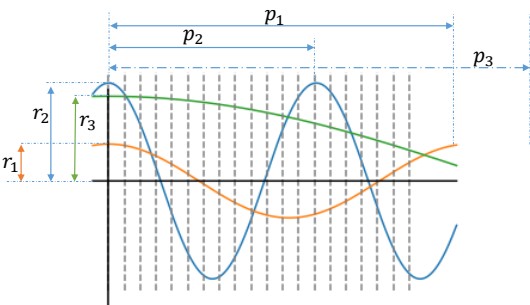

Figure 1: 3-dimensional complex embedding for a single word in different positions. The three wave functions (setting the initial phases as zero) show the real part of the embedding; the imaginary part has a $\frac{\pi}{2}$ phase difference and shows the same curves with its real-valued counterpart. The x-axis denotes the absolute position of a word and the y-axis denotes the value of each element in its word vector. Colours mark different dimensions of the embedding. The three cross points between the functions and each vertical line (corresponding to a specific position $pos$) represent the embedding for this word in the $pos$-th position.

can correlate with each other in a continuous function. Fig. 1 illustrates this type of word representation with a three-dimensional complex-valued embedding, where the amplitudes $\{r_1, r_2, r_3\}$ denote semantic aspects corresponding to classical word vectors, and periods $\{p_1, p_2, p_3\}$ denote how sensitive the word is to positional information. We further discuss the necessary properties of these functions to model sequential information and obtain a general solution in the form of a complex-valued embedding. Interestingly, there is a direct connection between a specific case of our general embedding and the well-known positional encoding in Vaswani et al. (2017) (see App. A).

We **contribute** (i) a novel paradigm that extends word vectors as continuous functions over changing variables like word position, and (ii) a general word embedding that models word order in a mathematically-sound manner. We integrate our complex word embeddings in state-of-the-art (SOTA) NN architectures (CNN, RNN, Transformer and experimentally find that it yields gains over both classical word embeddings and position-enriched word embeddings in text classification, machine translation and language modeling. Note that this is the first work in NLP to link imaginary numbers in complex-valued representation to concrete meanings (i.e., word order).

## 2 MODELLING WORD ORDER IN EMBEDDING SPACE

A Word Embedding (WE) generally defines a map $f_{we} : \mathbb{N} \to \mathbb{R}^D$ from a discrete **word** index to a $D$-dimensional real-valued vector and $\mathbb{N} = \{0, 1, 2, \ldots\}$. Similarly, a Position Embedding (PE) (Gehring et al., 2017; Vaswani et al., 2017) defines another map $f_{pe} : \mathbb{N} \to \mathbb{R}^D$ from a discrete **position** index to a vector. The final embedding for word $w_j$ ($w_j \in \mathbb{W}$ with index $j$ in a given vocabulary $\mathbb{W}$) in the $pos$-th position in a sentence is usually constructed by the sum

$$f(j, pos) = f_{we}(j) + f_{pe}(pos), \tag{1}$$

and $f(j, pos) \in \mathbb{R}^D$. Since both the word embedding map $f_w$ and the position embedding map $f_p$ only take integer values as word indexes or position indexes, embedding vectors for individual words or positions are trained independently. The independent training for each word vector is reasonable, since a word index is based on the order of a given arbitrary vocabulary and does not capture any specific sequential relationship with its neighboring words. However, the position index captures an ordered relationship, for instance adjacency or precedence, leading to the problem that position embeddings in individual positions (Gehring et al., 2017) are independent of each other; the ordered relationship between positions is not modelled. We refer to this as the *position independence problem*. This problem becomes more crucial when position embeddings are used in position-insensitive NNs, e.g., FastText (Mikolov et al., 2013b), ConvSeq (Gehring et al., 2017) and Transformer (Vaswani et al., 2017), because it is hard for such position-insensitive NNs with vanilla position embeddings (Gehring et al., 2017) to infer that $w_{j_1}$ in the $pos$-th position is close to $w_{j_2}$ in the $pos + 1$-th position, or that $w_{j_1}$ precedes $w_{j_2}$; instead, it is only inferred that $w_{j_1}$ and $w_{j_2}$ are in different positions, while the relative distance between them is almost unknown. Thus vanilla position embeddings (Gehring et al., 2017) cannot fully capture the sequential aspect of language.

Next, we first introduce the necessary properties to model word order in embeddings, and then give a unique solution to meet such properties.

## 2.1 Extending Vectors to Functions

In the general definition in Eq. 1, each dimension of the position embedding is obtained based on the discrete position indexes $\{0, 1, 2, ..., \text{pos}, ...\}$. This makes it difficult to model the ordered relationship between the positions. One solution to this problem is to build continuous functions over a variable (i.e., position index) to represent a specific word in an individual dimension. Formally, we define a general embedding as

$$f(j, \text{pos}) = \boldsymbol{g}_j(\text{pos}) \in \mathbb{R}^D, \tag{2}$$

where $\boldsymbol{g}_j$ is short for $\boldsymbol{g}_{we}(j) \in (\mathcal{F})^D$, indicating $D$ functions over position index pos, and $g_{we}(\cdot) : \mathbb{N} \to (\mathcal{F})^D$ is a mapping from a word index to $D$ functions. By expanding the $D$ dimension of $\boldsymbol{g}_j$, a word $w_j$ in the $pos$-th position can be represented as a $D$-dimensional vector as shown in

$$[g_{j,1}(\text{pos}), g_{j,2}(\text{pos}), ..., g_{j,D}(\text{pos})] \in \mathbb{R}^D, \tag{3}$$

in which $\forall g_{j,d}(\cdot) \in \mathcal{F} : \mathbb{N} \to \mathbb{R}, d \in \{1, 2, ..., D\}$ is a function over the position index $pos$. To move the word $w_j$ from the current position $pos$ to another one $pos'$, it needs only replace the variable $pos$ to $pos'$ without changing $\boldsymbol{g}_j$.

Functions for words, especially continuous functions, are expected to capture smooth transformation from a position to its adjacent position therefore modeling word order. The *position-independent* position embedding (Gehring et al., 2017) can be considered as a special case of our definition when it only takes independent values for individual positions in the embedding function.

## 2.2 Properties for the Functions to capture word order

Relative distance is hard to compute because position indices are not visible in NNs after vector embedding (discrete position indices are necessarily embedded as vectors like words to be back-propagated with the gradient). Hence, we claim that the modeling of relative distance in NNs should be *position-free*: absolute position indices cannot be directly accessed in intermediate layers. Instead of processing *position-free* operations in NNs to capture relative distance between words, prior work (Shaw et al., 2018; Dai et al., 2019) first calculates the relative distance between words, and then feeds the relative distance as an additional feature or as embeddings/weights to NNs, instead of directly feeding with the raw position indices.

Assume that words are embedded into $\mathbb{R}^D$, and let, for $1 \le d \le D$, the function $g_{j,d} : \mathbb{N} \to \mathbb{R}$ be the embedding function giving the $d$-th coordinate of the representation of word $w_j$ (i.e., $g_{j,d}(\text{pos})$ is the $d$-th coordinate of the embedding of $w_j$ if it occurs at position pos. In the following, we simply write $g$ instead of $g_{j,d}$ when there is no risk of confusion. Ideally, one would like there to exist a function $\text{Transform}_n : \mathbb{R} \to \mathbb{R}$ that transforms the embedding of any word at some position pos to the embedding of a word at position $\text{pos} + n$ such that $\text{Transform}_n$ is only dependent on the embedded value itself, but *independent* of the position pos, that is $\forall \text{pos} : g(\text{pos} + n) = \text{Transform}_n(g(\text{pos}))$.

Prior work in NLP (Li et al., 2019), Information Retrieval (Van Rijsbergen, 2004) and Machine Learning (Trabelsi et al., 2017) has shown the usefulness of complex numbers as richer representations. Complex word embeddings (Wang et al., 2019; Li et al., 2019; Li et al., 2018) have been used to model language. To investigate the potential of complex-valued representation, we extend the target domains of $g(\cdot)$ from $\mathbb{R}^D$ to $\mathbb{C}^D$ without losing generality, since real-valued numbers are specific complex numbers with their imaginary part being zero. This property regarding "position-free offset transformation" in complex-valued domains is formally defined in Property 1 below.

**Property 1. Position-free offset transformation**: An embedding function $g : \mathbb{N} \to \mathbb{C}$ is said to be a *position-free offset transformation* if there exists a function $\text{Transform} : \mathbb{N} \times \mathbb{C} \to \mathbb{C}$ (called the *witness*) such that for all $n \ge 1$, the function $\text{Transform}_n(\cdot) = \text{Transform}(n, \cdot)$ satisfies $\forall \text{pos} \in \mathbb{N} : g(\text{pos} + n) = \text{Transform}_n(g(\text{pos}))$. A position-free offset transformation $g$ is said to be *linearly witnessed* if there is a function $w : \mathbb{N} \to \mathbb{C}$ such that $g$ has a witness Transform satisfying, for all $n$, $\text{Transform}(n, \text{pos}) = \text{Transform}_n(\text{pos}) = w(n)$ (i.e., each $\text{Transform}_n$ is a linear function).

Additionally, a *boundedness* property is necessary to ensure that the position embedding can deal with text of any length ($pos$ could be large in a long document).

**Property 2. Boundedness**: The function over the variable position should be bounded, i.e. $\exists \delta \in \mathbb{R}^+, \forall \text{pos} \in \mathbb{N}, |g(\text{pos})| \le \delta$.

Formally, we prove the following claim that there is a unique solution that meets Properties 1 and 2 under the condition that the embedding function is linearly witnessed. We use linear functions because they are well-understood and simple with a single floating-point operation in NNs.

**Claim 1.** *A function $g : \mathbb{N} \to \mathbb{C}$ is a bounded and linearly witnessed position-free offset transformation iff it is on the form $g(\text{pos}) = z_2 z_1^{pos}$ for $z_1, z_2 \in \mathbb{C}$ with $|z_1| \leq 1$.*

*Proof.* Assume that $g$ is a bounded and linearly witnessed position-free offset transformation. Then, by linear witnessing, we have for all pos, $n_1, n_2 \in \mathbb{N}$:

$$w(n_1)w(n_2)g(\text{pos}) = w(n_2)g(\text{pos} + n_1) = g(\text{pos} + n_1 + n_2)$$
$$= \text{Transform}_{n_1+n_2}(g(\text{pos})) = w(n_1 + n_2)g(\text{pos})$$

whence $w(n_1+n_2) = w(n_1)w(n_2)$. Write $w(1) = z_1$ and $g(0) = z_2$. As $n_1, n_2 \in \mathbb{N}$ were arbitrary, we have $w(n) = (w(1))^n = z_1^n$ for all $n \in \mathbb{N}$. But then $g(\text{pos} + n) = w(n)g(\text{pos}) = z_1^n g(\text{pos})$. Furthermore, observe that for pos $\geq 1$, we have $g(\text{pos}) = g(1 + \text{pos} - 1) = w(\text{pos})g(0) = z_1^{\text{pos}} z_2 = z_2 z_1^{\text{pos}}$. For pos $= 0$, $g(0) = z_2 = z_2 z_1^0$, whence $g(\text{pos}) = z_2 z_1^{\text{pos}}$, as desired. Observe that if $|z_1| > 1$, then $g(\text{pos})$ is unbounded, whence we have $|z_1| \leq 1$. Conversely, assume that $g$ is on the form $g(\text{pos}) = z_2 z_1^{\text{pos}}$ with $|z_1| \leq 1$. Then, $|g(\text{pos})| \leq |z_2 z_1^{\text{pos}}| \leq |z_2||z_1^{\text{pos}}| \leq |z_2|$, whence $g$ is bounded. Define, for each $n \in \mathbb{N}$, $w(n) = z_1^n$ and $\text{Transform}_n(\text{pos}) = w(n)\text{pos}$. Then, for all pos, $n \in \mathbb{N}$,

$$g(\text{pos} + n) = z_2 z_1^{\text{pos}+n} = z_2 z_1^{\text{pos}} z_1^n = g(\text{pos})z_1^n = \text{Transform}_n(g(\text{pos}))$$

showing that $g$ is a linearly witnessed position-free offset transformation. $\square$

For any $z \in \mathbb{C}$, we may write $z = re^{i\theta} = r(\cos\theta + i\sin\theta)$. Thus, for the general form of the embedding $g$ from Theorem 1, we have:

$$g(\text{pos}) = z_2 z_1^{\text{pos}} = r_2 e^{i\theta_2}(r_1 e^{i\theta_1})^{\text{pos}} = r_2 r_1^{\text{pos}} e^{i(\theta_2 + \theta_1 \text{pos})} \text{ subject to } |r_1| \leq 1 \quad (4)$$

In implementations, the above definition of $g$ will lead to an optimization problem due to the constraint $|r_1| \leq 1$. A natural and simple way to avoid this is to fix $r_1 = 1$; note that $|e^{ix}| \equiv 1$, thus automatically satisfying the constraint, in contrast to a real-valued embedding where one would need to explicitly devise functions satisfying the constraint. Finally, Eq. 4 can be written in the simplified form: $g(\text{pos}) = re^{i(\omega \text{pos}+\theta)}$. Thus, one can think of $g$ as embedding positions counterclockwise on a complex circle of radius $r$ with a fixed period ($r$ is the amplitude term, $\theta$ is the initial phase term, $\frac{\omega}{2\pi}$ is the frequency, and $\frac{2\pi}{\omega}$ is the period term).

## 2.3 COMPLEX-VALUED WORD EMBEDDING

We now define our complex-valued word embedding $g$ as a map taking a word index $j$ and position word index pos to $\mathbb{C}^D$. For a word $w_j$ in position pos, our **general complex-valued embedding** is defined as $f(j, \text{pos}) = \boldsymbol{g}_j(\text{pos}) = \boldsymbol{r}_j e^{i(\boldsymbol{\omega}_j \text{pos} + \boldsymbol{\theta}_j)}$. Therefore, $f(j, \text{pos})$ is defined as:

$$[r_{j,1}e^{i(\omega_{j,1}\text{pos}+\theta_{j,1})}, ..., r_{j,2}e^{i(\omega_{j,2}\text{pos}+\theta_{j,2})}, \cdots, r_{j,D}e^{i(\omega_{j,D}\text{pos}+\theta_{j,D})}] \quad (5)$$

Note that each coordinate $d$ ($1 \leq d \leq D$) has a separate amplitude $r_{j,d}$, period $p_{j,d} = \frac{2\pi}{\omega_{j,d}}$, and initial phase $\theta_{j,d}$. In Fig. 1 each dimension is represented as a wave which is parameterized by an amplitude, a period/frequency, and an initial phase. The trainable parameters of the embedding are the amplitudes vector $\boldsymbol{r}_j = [r_{j,1}, ..., r_{j,D}]$, the period/frequency related weights $\boldsymbol{\omega}_j = [\omega_{j,1}, ..., \omega_{j,D}]$, and the initial phase vector $\boldsymbol{\theta}_j = [\theta_{j,1}, ..., \theta_{j,D}]$. Note that the mean values of $f(j, \cdot)$ over all positions are linearly dependent on the amplitude. Observe that the period/frequency determines to what degree the word is sensitive to the position. With an extremely long period (i.e., $\omega_j$ very small), the complex-valued embedding is approximately constant for all possible values of pos, and hence approximates a standard word embedding. Conversely, if the period is short, the embedding will be highly sensitive to the position argument.

In our embedding, the mean vectors of $f(j, \cdot)$ taken over all positions are linearly correlated to the amplitude embedding $\boldsymbol{r}_j = [r_{j,1}, ..., r_{j,K}]$ with a coefficient $\frac{2}{\pi}$. The amplitude $r_{j,d}$ of our embedding depends only on the word $w_j$ (and coordinate $d$), not on the position of the word, whence one can

| Dataset | train | test | vocab. | task | Classes |
|---------|-------|------|--------|------|---------|
| CR (Hu & Liu, 2014) | 4K | CV | 6K | product reviews | 2 |
| MPQA (Wiebe et al., 2005) | 11k | CV | 6K | opinion polarity | 2 |
| SUBJ (Pang & Lee, 2005) | 10k | CV | 21k | subjectivity | 2 |
| MR (Pang & Lee, 2005) | 11.9k | CV | 20k | movie reviews | 2 |
| SST (Socher et al., 2013) | 67k | 2.2k | 18k | movie reviews | 2 |
| TREC (Li & Roth, 2002) | 5.4k | 0.5k | 10k | Question | 6 |

Table 1: Dataset Statistics. CV means 10-fold cross validation. The last 2 datasets come with train/dev/test splits.

think of the vector $g_{pe}(j, \text{pos}) = [e^{i(\omega_{j,1}\text{pos}+\theta_{j,1})}, \cdots, e^{i(\omega_{j,D}\text{pos}+\theta_{j,D})}]$ as a "purely" positional embedding. Consequently, our complex embedding can be considered an element-wise multiplication between the word embedding $g_{we}(j) = [r_{j,1}, ..., r_{j,K}]$ and position embedding $g_{pe}$.

$$f(j, \text{pos}) = g_{we}(j) \odot g_{pe}(j, \text{pos}) \tag{6}$$

Prior work (Gehring et al., 2017; Vaswani et al., 2017) uses mean-weight addition between word embeddings $f_{we}$ and position embeddings $f_{pe}$ (all words share the weights). In our work, word embeddings and position embeddings are decoupled to some extent by element-wise multiplication and therefore the frequency/period terms (related to $\omega_{j,d}$) can adaptively adjust the importance between semantic and position information for each word and each dimension. In particular, with higher frequency (i.e., large $\omega_{j,d}$), the final embedding will change dramatically with the changing positions, while it can be fixed for any positions with an extremely-small frequency (i.e., small $\omega_{j,d}$). Interestingly, the well-known position embedding in Transformer (Vaswani et al., 2017) can be seen as a degraded version of one of our specific complex word embeddings (see the Appendix A).

## 3 EXPERIMENTAL EVALUATION

We evaluate our embeddings in text classification, machine translation and language modeling.

### 3.1 TEXT CLASSIFICATION

**Experimental Setup.** We use six popular text classification datasets: CR, MPQA, SUBJ, MR, SST, and TREC (see Tab. 1). We use accuracy as evaluation measure based on fixed train/dev/test splits or cross validation, as per prior work. We use Fasttext (Joulin et al., 2016), CNN (Kim, 2014), LSTM and Transformer (Vaswani et al., 2017) as NN baselines[2]. We use each of them: (1) without positional information; (2) with **Vanilla Position Embeddings (PE)** (randomly initialized and updated during training using the sum between word and position vectors (Gehring et al., 2017); (3) with **Trigonometric Position Embeddings (TPE)** (defining position embeddings as trigonometric functions as per Eq. 7); (4) with **Complex-vanilla** word embeddings (where the amplitude embedding is initialized by the pre-trained word vectors, and the phrase embedding is randomly initialized in a range from $-\pi$ to $\pi$ without considering word order (Wang et al., 2019)); and (5) with our order-aware complex-valued word embeddings, **Complex-order** (which encode position in the phase parts, train the periods, and where the amplitude embedding is also initialized by pretrained word vectors). For more details on the complex-valued extensions of NNs, see App. B and App. C.

Our embedding generally has $3 \times D \times |\mathbb{W}|$ parameters with D-dimensional word vectors and $|\mathbb{W}|$ words, while previous work (Mikolov et al., 2013b; Pennington et al., 2014) usually employs only $D \times |\mathbb{W}|$ parameters for embedding lookup tables. To increase efficiency and facilitate fair comparison with previous work we set initial phases $\boldsymbol{\theta}_j = [\theta_{j,1}, ..., \theta_{j,D}]$ to a shared constant value (such as zero). Furthermore, the period vectors $\omega_{j,d}$ depend on word index $j$ with length $|\mathbb{W}|$ and the coordinate index $d$ with length $D$. To decrease the number of parameters, one can either use a *word-sharing* scheme (i.e., $\omega_{j,d} = \omega_{.,d}$), or a *dimension-sharing* scheme ($\omega_{j,d} = \omega_{j,.}$), leading to $|\mathbb{W}| * D + |\mathbb{W}|$ and $|\mathbb{W}| * D + D$ parameters in total for the embedding layer.

---

[2]Graph convolutional networks (GCNs) (Beck et al., 2018; Sahu et al., 2019) also encode positional information. We do not compare against them because they encode positional information inherently as part of the model, which makes redundant any additional encoding of positional information at the embedding level.

Table 2: Text classification accuracy without position embeddings, with random position embeddings (PE), with trigonometric position embeddings (TPE), with complex-valued NNs without position embeddings (complex-vanilla), and with our complex-order embeddings. Superscripts §, †, ‡ and * mean a significant improvement over a baseline without position embeddings §, PE†, TPE‡ and Complex-vanilla * using Wilcoxon's signed-rank test $p<0.05$.

| Method | MR | SUBJ | CR | MPQA | SST | TREC |
|---|---|---|---|---|---|---|
| Fasttext | 0.765 | 0.916 | 0.789 | 0.874 | 0.788 | 0.874 |
| Fasttext-PE | 0.774 | 0.922 | 0.789 | 0.882 | 0.791 | 0.874 |
| Fasttext-TPE | 0.776 | 0.921 | 0.796 | 0.884 | 0.792 | 0.88 |
| Fasttext-Complex-vanilla | 0.773 | 0.918 | 0.79 | 0.867 | 0.803 | 0.872 |
| **Fasttext-Complex-order** | $\mathbf{0.787}^{\S\dagger\ddagger*}$ | $\mathbf{0.929}^{\S\dagger\ddagger*}$ | $\mathbf{0.800}^{\S\dagger\ddagger*}$ | $\mathbf{0.889}^{\S\dagger\ddagger*}$ | $\mathbf{0.809}^{\S\dagger\ddagger*}$ | $\mathbf{0.892}^{\S\dagger\ddagger*}$ |
| LSTM | 0.775 | 0.896 | 0.813 | 0.887 | 0.807 | 0.858 |
| LSTM-PE | 0.778 | 0.915 | 0.822 | 0.889 | 0.811 | 0.858 |
| LSTM-TPE | 0.776 | 0.912 | 0.814 | 0.888 | 0.813 | 0.865 |
| LSTM-Complex-vanilla | 0.765 | 0.907 | 0.810 | 0.823 | 0.784 | 0.784 |
| **LSTM-Complex-order** | $\mathbf{0.790}^{\S\dagger\ddagger*}$ | $\mathbf{0.926}^{\S\dagger\ddagger*}$ | $\mathbf{0.828}^{\S\dagger\ddagger*}$ | $\mathbf{0.897}^{\S\dagger\ddagger*}$ | $\mathbf{0.819}^{\S\dagger\ddagger*}$ | $\mathbf{0.869}^{\S\dagger\ddagger*}$ |
| CNN | 0.809 | 0.928 | 0.830 | 0.894 | 0.856 | 0.898 |
| CNN-PE | 0.816 | 0.938 | 0.831 | 0.897 | 0.856 | 0.890 |
| CNN-TPE | 0.815 | 0.938 | 0.836 | 0.896 | 0.838 | 0.918 |
| CNN-Complex-vanilla | 0.811 | 0.937 | 0.825 | 0.878 | 0.823 | 0.900 |
| **CNN-Complex-order** | $\mathbf{0.825}^{\S\dagger\ddagger*}$ | $\mathbf{0.951}^{\S\dagger\ddagger*}$ | $\mathbf{0.852}^{\S\dagger\ddagger*}$ | $\mathbf{0.906}^{\S\dagger\ddagger*}$ | $\mathbf{0.864}^{\S\dagger\ddagger*}$ | $\mathbf{0.939}^{\S\dagger\ddagger*}$ |
| Transformer w/o position embedding | 0.669 | 0.847 | 0.735 | 0.716 | 0.736 | 0.802 |
| Transformer-PE | 0.737 | 0.859 | 0.751 | 0.722 | 0.753 | 0.820 |
| Transformer-TPE (Vaswani et al., 2017) | 0.731 | 0.863 | 0.762 | 0.723 | 0.761 | 0.834 |
| Transformer-Complex-vanilla | 0.715 | 0.848 | 0.753 | 0.786 | 0.742 | 0.856 |
| **Transformer-Complex-order** | $\mathbf{0.746}^{\S\dagger\ddagger*}$ | $\mathbf{0.895}^{\S\dagger\ddagger*}$ | $\mathbf{0.806}^{\S\dagger\ddagger*}$ | $\mathbf{0.863}^{\S\dagger\ddagger*}$ | $\mathbf{0.813}^{\S\dagger\ddagger*}$ | $\mathbf{0.896}^{\S\dagger\ddagger*}$ |

Table 3: Text classification accuracy. ⋆ means that scores are reported from other papers.

| Method | MR | SUBJ | CR | MPQA | SST | TREC |
|---|---|---|---|---|---|---|
| Word2vec Bow (Conneau et al., 2017) ⋆ | 0.777 | 0.909 | 0.798 | 0.883 | 0.797 | 0.836 |
| Sent2Vec (Pagliardini et al., 2017) ⋆ | 0.763 | 0.912 | 0.791 | 0.872 | 0.802 | 0.858 |
| QuickThoughts (Logeswaran & Lee, 2018) ⋆ | 0.824 | 0.948 | 0.860 | 0.902 | - | 0.928 |
| InferSent (Conneau et al., 2017) ⋆ | 0.811 | 0.924 | **0.863** | 0.902 | 0.846 | 0.882 |
| QPDN (Wang et al., 2019) ⋆ | 0.801 | 0.927 | 0.810 | 0.870 | 0.839 | 0.882 |

We search the hyper parameters from a parameter pool, with batch size in $\{32, 64, 128\}$, learning rate in $\{0.001, 0.0001, 0.00001\}$, L2-regularization rate in $\{0, 0.001, 0.0001\}$, and number of hidden layer units in $\{120, 128\}$. We use pre-trained 300-dimensional vectors from word2vec (Mikolov et al., 2013a) in all models except for Transformers. The models with trainable trigonometric position embedding produce nearly identical results compared to the non-trainable version, therefore we report the result of fixed position embeddings as per Vaswani et al. (2017). We adopt narrow convolution and max pooling in CNN, with number of filters in $\{64, 128\}$, and size of filters in $\{3, 4, 5\}$. In all Transformer models, we only use the encoder layer to extract feature information, where the layer is 1, dimension of word and inner hidden are 256 and 512 respectively, and head number is 8.

**Results.** The results are shown in Tab. 2. Our complex-order embeddings outperform all other variations at all times. This gain in effectiveness comes at a negligible (or non-existent) cost in efficiency (it varies per NN architecture – see Fig. 2). CNNs are the best performing NN as expected following Bai et al. (2018). Tranformer NNs benefit the most from our complex-order embeddings, most likely because they are our weakest baseline. To contextualise these results, Tab. 3 shows classification accuracy of five typical approaches on the same datasets (as reported in the original papers). Our complex-order embeddings outperform all methods, except for the CR dataset, where InferSent is marginally better. Overall, our approach is on a par with the SOTA in embeddings.

We perform an ablation test (Tab. 4) on Transformer because it is the most common NN to be used with position embeddings. The two period-sharing schemas (dimension-sharing and word-sharing)

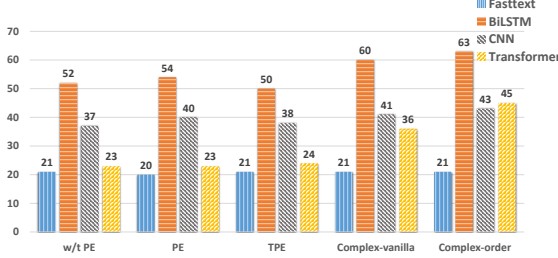

Figure 2: Computation time (seconds) per epoch in Tensorflow on TITAN X GPU.

Table 4: Ablation test for Transformer, showing the effect of (i) the definition of embedding layer($f_d(j, \text{pos})$), and (ii) whether the real-part and imaginary transition share the weights, i.e., $\Re(W^{Q/K/V}) = \Im(W^{Q/K/V})$.

| Method | Setting | | Params | Accuracy | $\Delta$ |
|---|---|---|---|---|---|
| | $f_d(j, \text{pos})$ | share in $W^{Q/K/V}$ | | | |
| Transformer-complex-order | $r_{j,d}e^{i(\omega_{j,d}\text{pos})}$ | $\times$ | 8.33M | **0.813** | - |
| adding initial phases | $r_{j,d}e^{i(\omega_{j,d}\text{pos}+\theta_{j,d})}$ | $\times$ | 11.89M | 0.785 | -0.028 |
| dimension-sharing period schema | $r_{j,d}e^{i\omega_{j,\cdot}\text{pos}}$ | $\times$ | 5.82M | 0.797 | -0.016 |
| word-sharing period schema | $r_{j,d}e^{i\omega_{\cdot,d}\text{pos}}$ | $\times$ | 5.81M | 0.805 | -0.008 |
| dimension-sharing amplitude schema | $r_{j,\cdot}e^{i\omega_{j,\cdot}\text{pos}}$ | $\times$ | 5.82M | 0.798 | -0.015 |
| word-sharing amplitude schema | $r_{\cdot,d}e^{i\omega_{\cdot,d}\text{pos}}$ | $\times$ | 5.81M | 0.804 | -0.009 |
| w/t encoding positions (complex-vanilla) | $r_{j,d}e^{i\omega_{j,d}}$ | $\times$ | 9.38M | 0.764 | -0.049 |
| dimension-sharing period schema | $r_{j,d}e^{i\omega_{j,\cdot}\text{pos}}$ | $\checkmark$ | 4.77M | 0.794 | -0.019 |
| word-sharing period schema | $r_{j,d}e^{i\omega_{\cdot,d}\text{pos}}$ | $\checkmark$ | 4.76M | 0.797 | -0.016 |
| dimension-sharing amplitude schema | $r_{j,\cdot}e^{i\omega_{j,\cdot}\text{pos}}$ | $\checkmark$ | 4.77M | 0.792 | -0.021 |
| word-sharing amplitude schema | $r_{\cdot,d}e^{i\omega_{\cdot,d}\text{pos}}$ | $\checkmark$ | 4.76M | 0.801 | -0.012 |
| w/t encoding positions (complex-vanilla) | $r_{j,d}e^{i\omega_{j,d}}$ | $\checkmark$ | 8.33M | 0.743 | -0.07 |
| vanilla Transformer (Vaswani et al., 2017) | $WE_{j,d} + PE_d$ | - | 4.1M | 0.761 | -0.052 |

slightly drop performance, because fewer parameters limit the representative power. Adding initial phases also hurts performance, although we observed that the loss could decrease faster in early epochs compared to the setting without offset. The negative effect of initial phases may be due to periodicity, and $\omega$ cannot be directly regularized with L2-norm penalties. The sharing schemes slightly decrease the performance with less parameters. More details of the learned periods/frequencies (e.g. the distributions of periods/frequencies and case studies) are shown in App. D.

Note that the word-sharing schema outperform the Vanilla Transformer, (both have a comparable number of parameters). If we choose $\Re(W^{Q/K/V}) = \Im(W^{Q/K/V})$, the additional parameters in the embedding layers will affect much less the whole parameter scale in the multiple-layer Transformer, since a embedding layer is only used in the first layer instead of the following Transformer layers.

## 3.2 MACHINE TRANSLATION

**Experimental Setup.** We use the standard WMT 2016 English-German dataset (Sennrich et al., 2016), whose training set consists of 29,000 sentence pairs. We use four baselines: basic Attentional encoder-decoder (AED) (Bahdanau et al., 2014); AED with Byte-pair encoding (BPE) subword segmentation for open-vocabulary translation (Sennrich et al., 2016); AED with extra linguistic features (morphological, part-of-speech, and syntactic dependency labels) (Sennrich & Haddow, 2016); and a 6-layer Transformer. Our approach (Transformer Complex-order) uses a batch size of 64, a head of 8, 6 layers, the rate of dropout is 0.1, and the dimension of the word embedding is 512. The embedding layer does not use initial phases, i.e., following $f(j, pos) = \boldsymbol{r}_j e^{i(\boldsymbol{\omega}_j pos)}$. We evaluate MT performance with the Bilingual Evaluation Understudy (BLEU) measure.

Table 5: Machine translation results. ⋆ marks scores reported from other papers.

| Method | BLEU |
|---|---|
| AED (Bahdanau et al., 2014) ⋆ | 26.8 |
| AED+Linguistic (Sennrich & Haddow, 2016) ⋆ | 28.4 |
| AED+BPE (Sennrich et al., 2016) ⋆ | 34.2 |
| Transformer (Ma et al., 2019) ⋆ | 34.5 |
| Transformer complex vanilla | 34.7 |
| **Transformer Complex-order** | **35.8** |

Table 6: Language modeling results. ⋆ marks scores reported from other papers.

| Method | BPC |
|---|---|
| BN-LSTM (Cooijmans et al., 2016) ⋆ | 1.36 |
| LN HM-LSTM (Chung et al., 2016) ⋆ | 1.29 |
| RHN (Zilly et al., 2017) ⋆ | 1.27 |
| Large mLSTM (Krause et al., 2016) ⋆ | 1.27 |
| Transformer XL 6L (Dai et al., 2019) | 1.29 |
| Transformer complex vanilla | 1.30 |
| **Transformer XL Complex-order 6L** | **1.26** |

**Results** Tab. 5 shows the MT results. Our approach outperforms all baselines. Two things are worth noting: (1) Both the vanilla Transformer and our Transformer Complex-order outperform the three Attentional encoder-decoder baselines which are based on an LSTM encoder and decoder, even when AED uses additional features. (2) Our Transformer Complex-Order outperforms the Vanilla Transformer and complex-vanilla Transformer by 1.3 and 1.1 in absolute BLEU score respectively.

## 3.3 LANGUAGE MODELING

**Experimental Setup.** We use the text8 (Mahoney, 2011) dataset, consisting of English Wikipedia articles. The text is lowercased from a to z, and space. The dataset contains 100M characters (90M for training, 5M for dev, and 5M for testing, as per Mikolov et al. (2012)). We use as baselines BN-LSTM, LN HM-LSTM RHN and Large mLSTM, which are typical recurrent NNs for language modeling in this dataset. We evaluate performance with the Bits Per Character (BPC) measure, (the lower, the better). We run the coder in Dai et al. (2019) with 6 layers for Transformer XL 6L; our model, named Transformer XL complex-order, directly replaces the word embedding with our proposed embedding under the same setting. We choose 6 layers due to limitations in computing resources. For Transformer XL Complex-order, all other parameter settings are as for Transformer XL (Dai et al., 2019). Our complex-order model does not use initial phases.

**Results.** We see in Tab. 6 that our method outperforms all baselines. The first four baselines (BN-LSTM, LN HM-LSTM, RHN and Large mLSTM) are based on recurrent NNs and rely on different regulation methods to become applicable in multiple-layer recurrent architectures. Transformer-based architectures can easily be stacked with multiple layers due to their advantages in parallelization, however the vanilla Transformer does not outperform the multiple-layer recurrent baselines, most likely due to its limitation of 6 layers. Our Transformer XL Complex-order outperforms its vanilla counterpart under the 6-layer setting (and also strong recurrent network baselines), demonstrating that our embedding also generalizes well in tasks with long-term dependency. With limited resources, slightly increasing the parameters in the feature layer like our proposed embedding could be more beneficial than stacking more layers with linearly increasing parameters.

## 4 RELATED WORK

Complex-valued NNs are not new (Georgiou & Koutsougeras, 1992; Kim & Adalı, 2003; Hirose, 2003). Complex-valued weights have been used in NNs, motivated by biology (Reichert & Serre, 2013), and also as signal processing in speech recognition (Shi et al., 2006). More recently, Arjovsky et al. (2016) shifted RNNs into the complex domain and Wolter & Yao (2018) proposed a novel complex gated recurrent cell. Trabelsi et al. (2017) also developed a complex-valued NN for computer vision and audio processing.

Complex numbers have also been applied to text processing like (Van Rijsbergen, 2004; Melucci, 2015; Blacoe et al., 2013). Trouillon et al. (2016) adopt complex embedding for entities in Knowledge Graph Completion to represent antisymmetric relations with Hermitian dot product. Li et al. (2019); Wang et al. (2019) extend word embeddings to complex-valued fashion in quantum probability driven NNs, seeing the overview in Wang et al. (2019). However, the physical meaning of both the complex-valued entity and word embeddings is unknown, since a complex number was considered as two real numbers in a black-box learning paradigm. Our work first links the phase in complex numbers to word position to define concrete physical meaning in document representations.

## 5 Conclusions

We extended word vectors to word functions with a variable i.e. position, to model the smooth shift among sequential word positions and therefore implicitly capture relative distances between words. These functions are well-defined to model the relative distances, therefore we derive a general solution in complex-valued fashion. Interestingly, the position embedding in Vaswani et al. (2017) can be considered a simplified version of our approach. We extend CNN, RNN and Transformer NNs to complex-valued versions to incorporate our complex embedding. Experiments on text classification, machine translation and language modeling show that our embeddings are more effective than vanilla position embeddings (Vaswani et al., 2017).

### Acknowledgments

We thank Massimo Melucci and Emanuele Di Buccio for their helpful comments, Xindian Ma for his detailed experimental suggestions.

This work is supported by the Quantum Access and Retrieval Theory (QUARTZ) project, which has received funding from the European Union's Horizon 2020 research and innovation programme under the Marie Skłodowska-Curie grant agreement No. 721321. Peng Zhang and Donghao Zhang are supported in part by Natural Science Foundation of China (grant No. 61772363, U1636203)

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

## A LINKING TO THE POSITION EMBEDDINGS IN (VASWANI ET AL., 2017)

Vaswani et al. (2017) proposed a new initialization for position embedding, resulting in comparable performance with previous one (Gehring et al., 2017) even without fine-tuning. The position embedding is empirically selected as

$$PE_{2k}(\cdot, pos) = \sin(pos/10000^{2k/d_{model}})$$
$$PE_{2k+1}(\cdot, pos) = \cos(pos/10000^{2k/d_{model}})$$

$$(7)$$

Where $pos$ is the position index, $2k$ and $2k+1$ is the dimension index and $d_{model}$ is the dimension size of embedding. The reason for choosing this position embedding was not well-explained and its general extension is unknown, leading to some difficulties to improve it.

We claim that the proposed position embedding in (Vaswani et al., 2017) is a degraded version of one of our specific complex word embedding in word-sharing schema (i.e., $\omega_{j,d} = \omega_{\cdot,d}$), in which $p_{j,k} = 2\pi \times 10000^{2k/d_{model}}$ and the initial phases are set as zero. In our complex-valued position embedding, let $f_{pe,k}(\cdot, pos) = e^{i \times 10000^{2k/d_{model}}} = \cos(10000^{2k/d_{model}} pos) + i\sin(10000^{2k/d_{model}} pos)$. Note that there exists a bi-jection between $PE(\cdot, pos)$ and $f_{pe,k}(\cdot, pos)$:

$$PE_{2k}(\cdot, pos) = \Im(f_{pe,k}(\cdot, pos)),$$
$$PE_{2k+1}(\cdot, pos) = \Re(f_{pe,k}(\cdot, pos))$$

$$(8)$$

where $\Re$ and $\Im$ are the operations to take the real and imaginary part of a complex-valued number. Its inverse transformation is

$$f_{pe,k}(\cdot, pos) = PE_{2k+1}(\cdot, pos) + iPE_{2k}(\cdot, pos)$$

$$(9)$$

In our overall embedding, each dimension $f_k(j, pos) = f_{we,k}(j) \odot f_{pe,k}(\cdot, pos)$ in our approaches, while it is $E_k(j, pos) = WE_k(j) + PE_k(\cdot, pos)$ in (Gehring et al., 2017; Vaswani et al., 2017). Hence the position embedding in (Vaswani et al., 2017) is equivalent, albeit not identical, to our complex-valued position embedding with $p_{j,k} = 2\pi \times 10000^{2k/d_{model}}$. It is therefore a particular case of our complex-valued position embedding, the word-sharing schema in which all words share the same period at a certain dimension, i.e, $p_{j,k} = p_{\cdot,k}$ is irrelevant to the choice of $j$.

## B INTEGRATING COMPLEX-VALUED EMBEDDING TO GENERAL NEURAL NETWORKS

Neural networks are typically given real numbers as inputs and return real numbers as outputs. To accommodate complex numbers as in- and output, we devise a complex-valued version of various neural network layers i.e. complex-valued FastText with dense layer, CNN, and RNN. Unlike existing complex-valued neural networks (Trabelsi et al., 2017; Wolter & Yao, 2018), our feature layers are also converted into complex-valued layers.

**Complex-valued FastText** FastTest (Joulin et al., 2016) is a simple and efficient neural network architecture using a dense layer over the sum of all word embeddings for general text classification. For a linear dense layer, i.e., $z = \text{dense}(x + iy)$, where $x + iy$ and $z$ denote the complex-valued in- and output, respectively. Let $W = A + iB$ and $b = c + id$ be complex-valued linear weights and bias, respectively. Then, the complex-valued dense layer is given by:

$$z = \sigma(Ax - By + c) + i\sigma(Bx + Ay + d)$$

$$(10)$$

where $\sigma$ is a real-valued activation function such as the sigmoid function. By rewriting (10) in matrix form (Trabelsi et al., 2017), we obtain:

$$\begin{bmatrix} \Re(z) \\ \Im(z) \end{bmatrix} = \begin{bmatrix} \sigma(Ax - By + c) \\ \sigma(Bx + Ay + d) \end{bmatrix}$$

$$(11)$$

where, for $z = x + iy$, $\Re(z) = x$ and $\Im(z) = y$. To save parameters and fairly compare with our real-valued baselines, the weights for real-part and imaginary-part input can be shared, i.e., $A = B, c = d$.

**Complex-valued CNN** For the complex-valued version of the convolution operation Trabelsi et al. (2017), we similarly define a complex-valued convolution with separate real and imaginary kernels $\boldsymbol{A}$ and $\boldsymbol{B}$, to compute convolutions on the real and imaginary parts of the input in Eq. 10. A complex-valued CNN network is constructed by stacking the operations based on a complex-valued convolution kernel, adding a complex dense layer in (10), and taking their norm as the final prediction in the last layer.

**Complex-valued RNN** The basic complex RNN formulation is:

$$h_t^C = f\left(\boldsymbol{W}^h \boldsymbol{h}_{t-1} + \boldsymbol{W}^z \boldsymbol{z}_t + \boldsymbol{b}\right) \tag{12}$$

where $\boldsymbol{z}_t$ and $\boldsymbol{h}_t$ represent the complex-valued input and complex-value hidden state vectors at time $t$, $\boldsymbol{b}$ is a complex-valued bias, $\boldsymbol{W}^h$ and $\boldsymbol{W}^z$ are complex-valued weight transitions for hidden state and input state, and $f(\boldsymbol{z}) = \sigma\left(\Re\left(\boldsymbol{z}\right)\right) + i\sigma\left(\Im\left(\boldsymbol{z}\right)\right)$ is the activation function. The multiplication $\boldsymbol{W}^h \boldsymbol{h}_{t-1}$ and $\boldsymbol{W}^z \boldsymbol{z}_t$ is computed as defined in (10) above. Similarly, the complex-valued gates are used in LSTM via operations as in (12). In the final layer, a $l2$-norm operation is adopted to obtain a real-valued loss for backpropagation.

**Complex-valued Transformer** The main components in the Transformer are self-attention sublayers and position-wise feed-forward (FFN) sublayers. A self-attention sublayer employs $h$ attention heads and the concatenation of all heads is used as the output followed by a parameterized linear transformation. For a sequence embedded as complex-valued vector $input = \{\boldsymbol{w}_1, \boldsymbol{w}_2, ..., \boldsymbol{w}_n\}$, the output of each head is computed as a weighted sum of a linear transformation of the input sequence itself, namely

$$output_i = \sum_j a_{i,j} \boldsymbol{w}_j \boldsymbol{W}^V, \tag{13}$$

where $\boldsymbol{w}_j$ is a complex-valued vector and $\boldsymbol{W}^V$ is a complex linear transformation; therefore $output_i$ is also complex. Hence, $output$ is a sequence of complex-valued vectors with the same shape as $input$. The weight coefficient, $a_{i,j}$, which is defined as in real-valued domain, is calculated as the softmax of the product between complex-valued query vectors and key vectors: $a_{i,j} = \text{softmax} \frac{e_{i,j}}{\sum_{i=1}^n e_{i,j}}$, and

$$e_{i,j} = \sqrt{\frac{\Re(z)^2 + \Im(z)^2}{n}}, z = \left(\boldsymbol{w}_i \boldsymbol{W}^Q\right)\left(\boldsymbol{w}_j \boldsymbol{W}^K\right)^\dagger, \tag{14}$$

z is a complex number since $\boldsymbol{w}_i$ and $\boldsymbol{w}_j$ are complex-valued vectors and $\boldsymbol{W}^Q, \boldsymbol{W}^K$ are complex-valued transformation. To extend another variant of Transformer called Transformer XL, we keep its original relative position embedding and additionally replace its word embedding with our proposed embedding.

Correspondingly, the FFN sublayer can easily be extended to a complex-valued version by replacing the real-valued layers with complex-valued ones. We use batch-normalization separately for the real and imaginary parts.

## C    IMPLEMENTATION OF THE PROPOSED EMBEDDING

Words functions are implemented in neural networks by storing the function parameters $\{\boldsymbol{r}, \boldsymbol{\omega}, \boldsymbol{\theta}\}$ and then construct the values based on the arguments. Based on the definition, the implementation of the proposed embedding can easily be implemented with only modifying the embedding layer. We list the basic code to construct our general embedding as below:

```
import torch
import math
class ComplexNN(torch.nn.Module):
    def __init__(self, opt):
        super(ComplexNN, self).__init__()
        self.word_emb = torch.nn.Embedding(opt.n_token, opt.d_model)
        self.frequency_emb = torch.nn.Embedding(opt.n_token, opt.d_model)
        self.initial_phase_emb = torch.nn.Embedding(opt.n_token, opt.d_model)
```

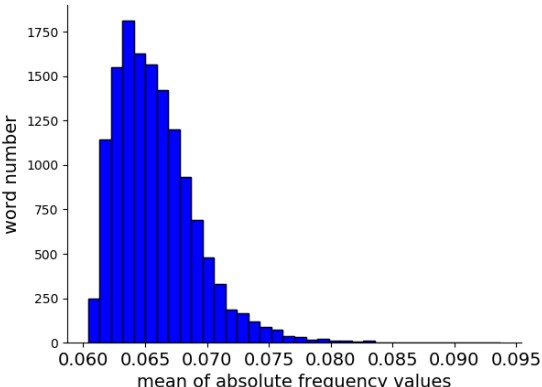

Figure 3: The distribution of the $\delta_j$. Higher values mean that the word representations are more sensitive to the word positions.

```python
def get_embedding(self, x):

    amplitude = self.word_emb(x)
    frequency = self.frequency_emb(x)
    self.initial_phase_emb.weight = torch.nn.Parameter(self.initial_phase_emb.weight
        % (2 * math.pi))

    sent_len=x.size(-1)
    pos_seq = torch.arange(1,  sent_len + 1, 1.0,device=amplitude.device)

    pos_seq = pos_seq.unsqueeze(0).unsqueeze(-1)
    pos_seq = pos_seq.repeat([x.size(0),1,amplitude.size(-1)])

    dimension_bais = self.initial_phase_emb (x)

    enc_output_phase = torch.mul(pos_seq,frequency)+ dimension_bais
    enc_output_real = amplitude * torch.cos(enc_output_phase)
    enc_output_image = amplitude * torch.sin(enc_output_phase)
    # return torch.cat([enc_output_real,enc_output_image],-1)
    return enc_output_real,enc_output_image

def forward(self, x) :
    return self.get_embedding(x)
...
```

Note that both the frequency vectors $\omega$ and initial-phase vectors $\theta$ can be shared between words or dimensions, to save parameters. The proposed embedding can be also used in real-valued neural networks if one directly concatenates the real-part numbers and imaginary-part numbers as a double-size real-valued vector; therefore it could easily be extended in any existing networks without any complex-valued components. For instance, it could be a good extension for Transformer based pretrained models like (Devlin et al., 2018) by enriching the feature layer.

## D    VISUALIZATION OF FREQUENCIES/PERIODS

After training, we obtain the frequency vector $\omega$ for each word. For each word, the mean value of the absolute frequency values, i.e., $\delta_j = \frac{1}{|D|} \sum_{d=1}^{D} |\omega_{j,d}|$ is considered as a metric to test the positional sensitivities of the word, since a period value could be negative during training. The density of the $\delta_j$ is shown in Fig. 3.

Words with the 50 greatest, and the 50 smallest, frequencies in the SST dataset are shown in Tab. 7. For the words with greatest frequencies, most of them are strong sentiment words like "worst"

,"stupid" and "powerful"; a reason for this may be that such words appear in many positions in many documents during training, and thus they are more sensitive to the positions. Conversely, there are fewer words expressing strong sentiment among words with smaller frequencies, as shown in the second row.

| | words |
|---|---|
| greatest frequencies in descending order | **worst** solid **stupid powerful** mess **wonderful remarkable** suffers intoxicating **thoughtful rare** captures portrait gem frontal **terrific unique** wannabe **witty lousy pointless** contrived none **worse refreshingly charming** inventive **amazing junk** incoherent refreshing mediocre **unfunny** thinks **enjoyed heartbreaking delightfully** crisp **brilliant** heart spirit **perfectly** nowhere mistake **engrossing fashioned excellent unexpected wonderfully** means |
| smallest frequencies in ascending order | slowly proposal schemes roiling juliette titles fabric superstar ah wow choreographed **tastelessness** beg **fabulous** muccino jacobi legendary jae rate example code sensation counter deaths hall eun drug mctiernan storylines cellophane wild motion ups trick comedy entertained mission **frightening** witnesses snoots liners african groan satisfaction calm saturday estranged holm refuses **inquisitive** |

Table 7: Words with greatest frequencies and frequencies periods (based on $\delta_j$) in SST (a sentiment classification task), all words are converted to lower-case. The strong sentiment words are bold based on manual labeling.

