# OpenReview forum: "Encoding word order in complex embeddings"
_ICLR.cc/2020/Conference — Accept (Spotlight)_

### Official Review · AnonReviewer2 · 2019-10-20
**Official Blind Review #2**

**Rating:** 6

**Review:**

### Summary

The authors present a "natural" way of encoding position information into word embeddings and present extensive empirical evidence to support their method. I believe that paper meets the bar for acceptance.

### Details

The paper "Encoding word order in complex embeddings" presents a method for making word embeddings position dependent. The idea in a nutshell is to map each discrete position , n, to a value  `   A exp( freq_{word, dim} × n )` . So a word embedding is a collection of complex valued signals sampled at discrete points.

The frequency is dependent on each word and each dimension in general. The authors motivate/justify this particular formulation via their Claim 1, which argues that their particular formulation uniquely satisfies two intuitive constraints. Although one of those constraints (i.e. linearly witnessed Position-free offset) almost completely specifies the solution.

The experiments in the paper are fairly thorough and cover text classification, machine translation and language modeling. Through the comparative experiments the complex embeddings we can see that the formulation in this paper outperforms existing SOTA methods, sometimes with significantly difference such as a difference of 1.3 BLEU point on the MT task.

I would have liked to say that the ablation are similarly conclusive but there seem to be a problem in the table, which eroded my confidence:

1. The number of parameters in rows 5 and 8  (w/t encoding positions, share / not-share respectively) are reported to be 9.38M and 8.33M which has to be wrong. Similar problem happens with other pairs. And now I am not sure whether the results were also swapped or not. Still the results in general trend in the right direction.

### Possible improvements to the paper

1. The main weakness of the paper is that the authors repeatedly mention that encoding the position as a multiplicative factor which is multiplied to the frequency gives leads to a more decoupled/interpretable embedding but their experiments are solely focused on accuracy measurement. I would have liked to see the authors carry out more experiments to see whether the frequency parameters really are interpretable? For example,
      -  What is the histogram of the frequencies ? Are some of them negative?
      - Which word has the highest frequencies (pooled over all dimensions) in absolute term? Does it make sense that that word's meaning is so position dependent? For example, positions can capture subjects versus objects in english, but they will more reliably reflect the subject versus verb distinction in hindi.
      - Are the word frequencies by themselves predictive of anything? For example, what happens if the word embedding amplitudes are tied across words or dimensions? We expect the performance to be bad but how bad?

These kinds of ablations  / qualitative analysis will really make the paper more informative and interesting. Right now it just seems like yet another paper where the capacity of the model is increased and the accuracy increases. Specially because the delta improvement over the fixed positional embeddings of (Transformer-TPE Vaswani et al. 2017) is so limited.



### Edit after the author response

The authors have made the required corrections and added the necessary analysis. One interesting outcome was that the "word-sharing amplitude schema" seems to drop so little in performance, it's almost like all the information can be coded in just the phase vectors alone. It will be nice if the authors release their trained phase embeddings as well, for the words.

**Experience Assessment:**

I have published one or two papers in this area.

**Review Assessment: Checking Correctness Of Derivations And Theory:**

I carefully checked the derivations and theory.

**Review Assessment: Checking Correctness Of Experiments:**

I carefully checked the experiments.

**Review Assessment: Thoroughness In Paper Reading:**

I read the paper thoroughly.

---

> ### Author Response · Authors · 2019-11-14
> **To Reviewer #2**
>
> Thank you for your review.
>
> There was a typo in the ablation table (Table 4).  We have swapped the notations for whether sharing the W for QKV transformations (in the third column, the sharing notation (×) and non-sharing notation (√)).
>
> There are indeed some negative values for frequencies/periods. Thus we define the overall frequency as the mean of the absolute values, and draw a histogram of the overall frequencies (Fig. 3). The word frequencies could be an indicator of whether the word representation is sensitive to where (which position) the word appears; the higher frequency the word has, the more sensitive the word representation is to the position. We have added appendix D, where we show that the most frequent words typically express strong sentiment.
>
> We added to the paper an ablation test (dimension-sharing amplitude and word-sharing amplitude schemas) where embedding amplitudes are tied across words or dimensions in Table 4 (and a brief discussion).

---

### Official Review · AnonReviewer1 · 2019-10-23
**Official Blind Review #1**

**Rating:** 8

**Review:**

This paper proposes to learn position varying embeddings of words using complex numbers. Specifically, this work learns a position embedding by learning a continuous function that respects relative position based constraints and is bounded. The authors show that complex representations are an ideal fit for this purpose wherein the amplitude of the complex wave represents the base word embedding that is positionally invariants and the "wave" part encodes encodes the evolution of each dimension with position  as a periodic function with a learnable phase and period.

Results are shown on text classification baselines and improvements are small. In the experiments related to machine translation and language modeling, some relevant baselines are missing (which were covered in the text classification case) , most importantly, Vaswani etal 2017 variant of complex position embeddings and "complex-vanilla", and all the numbers for other baselines are reported from the corresponding papers, hence it is unclear whether the improvement shown is strictly comparable or not.

Moreover, a question that is unanswered is how does the periodicity affect the quality of embeddings. Basically, because of the periodic nature, the dimensions will take the same value for multiple positions spaced out according to the period. Is this a good assumption? Now, with large periods, for a finite practical length value, the periodic effects might end up not being observed but is that the case in the models that this approach learns? It would be great if authors could characterize the contexts/ words for which the periods are small and the contexts for which they are large. Basically, my concern is about the effect of getting the same embedding as output for different positions which is very likely if most of the periods learnt are small.

Also, empirical results with some other functions (maybe unbounded, or non-linear functions that do not respect relative positional constraints) would be insightful in order to assess the need for the desiderata laid out for the position sensitive functions. Finally, the "iff" proof needs to be cleaned up because I am not still not convinced if the proof holds oin both the directions and I believe there could be other functions with desired properties.

There are minor typos in equations like last line of "Property 1" related to g_pos, x.


**Experience Assessment:**

I have published one or two papers in this area.

**Review Assessment: Checking Correctness Of Derivations And Theory:**

I carefully checked the derivations and theory.

**Review Assessment: Checking Correctness Of Experiments:**

I carefully checked the experiments.

**Review Assessment: Thoroughness In Paper Reading:**

I read the paper thoroughly.

---

> ### Author Response · Authors · 2019-11-14
> **To Reviewer # 1**
>
> Thank you for your review.
>
> We have added experiments with one more baseline (complex Vanilla) in the revised paper (Table 5.1 and 5.2). The reason why we did not add the remaining 2 baselines to Table 5.1 and 5.2 is time limitations within this rebuttal period. Note however that prior work (Vaswani et al. 2017) states that trainable PE and TPE have nearly identical results (our experiments on text classification tasks also show the same trend). Thus we include only TPE in Table 5.1 and 5.2, which is the standard Transformer implementation
>
> Regarding whether the improvements are strictly comparable or not, we ran all baselines using code from SOTA publications [2,3] and changed the embedding as proposed in our paper without modifying the baselines in any other way. We have released our code: https://github.com/iclr-complex-order/complex-order .
>
> Regarding the third comment, we have now added an appendix D where we characterize the contexts for which the periods are small and the contexts for which they are large.
>
> We agree that experimenting with functions that do not satisfy both of our desired properties would be interesting, and we intend to do so in future work.
>
> We have reviewed and clarified the proof of the ‘iff’ condition.
>
> We have fixed the typo in the last line of Property 1.

---

> > ### Comment · AnonReviewer1 · 2019-11-14
> > **Addresses many of my concerns**
> >
> > The authors address a lot of my concerns and I am leaning toward accepting this paper at ICLR. Although, some interesting analysis is presented in Appendix D, I still feel that a more in depth quantitative and qualitative analysis of the positional embeddings and the learned frequencies should be a part of the main paper. From the appendix, it seems like simply the frequency of word occurrence might be correlated with the learned positional embeddings . An in depth analysis of position sensitivity would help with a clearer understanding of the contributions of the proposed approach.

---

### Official Review · AnonReviewer4 · 2019-10-28
**Official Blind Review #4**

**Rating:** 6

**Review:**


### Problem and Previous Research
This paper tackles the problem of incorporating the sequential structure of words for text processing.
Previous research [Gehring et al., ICML'17; Vaswani et al., NeurIPS'17] tackles the problem by adding position embeddings at the feature level.
Supported by recent empirical results [Shaw et al., NAACL'18; Dai et al., ACL'19], the paper argues that these position embeddings are independent i.e. do not consider relations between neighbouring word positions.

### Contributions
To address this key limitation of previous research, the paper proposes to define the embedding of each word through a continuous function over its position (so that embeddings shift smoothly with increasing positions thereby modelling word order).
The paper then lists two properties that such a function needs to satisfy and proposes to use the complex space as the target domain of the function.
Experimental results on text classification, machine translation, and language modelling show gains over classical and position-enriched word embeddings.

### Pros and Cons
Overall, the paper tackles an important problem in word embeddings and proposes a principled approach to the problem.
However, the paper could be further strengthened by positioning itself with respect to other existing neural network-based approaches that incorporate sequential structure e.g. graph neural networks (GNNs).
Gated-graph neural networks (GGNN) [Beck et al., ACL'18] and graph convolutional networks [Sahu et al., ACL'19] use GNNs on graphs with words as nodes and labelled edges (adjacence, precedence, etc.) between nodes to model the sequential structure between words.

### Possible Improvements
On the empirical side, GGNN of Beck et al. seems esp. relevant since they show improvements on machine translation.
A GNN-based baseline to compare against is to use Transformer - Complex - vanilla embeddings as features to a GGNN on the graph with words as nodes and adjacence, precedence labelled edges between neighbouring words.
[Beck et al., ACL'18] Graph-to-Sequence Learning using Gated Graph Neural Networks
[Sahu et al., ACL'19] Inter-sentence Relation Extraction with Document-level Graph Convolutional Neural Network

I am open to revising my rating based on the responses of the authors.

**Experience Assessment:**

I have read many papers in this area.

**Review Assessment: Checking Correctness Of Derivations And Theory:**

I carefully checked the derivations and theory.

**Review Assessment: Checking Correctness Of Experiments:**

I assessed the sensibility of the experiments.

**Review Assessment: Thoroughness In Paper Reading:**

I read the paper at least twice and used my best judgement in assessing the paper.

---

> ### Comment · AnonReviewer3 · 2019-11-14
> **Novelty**
>
> I'm not sure the authors responded to this review, but I wanted to champion this paper as I found it exceptional.
>
> I found the review AnonReviewer4 a little bit severe and I'm not sure I agree that the experiments about using GNNs would add anything to the paper:
> - first, sequences are a special type of GNN, and a very common one that justifies a publication if we show improvement for important sequence problems.
> - I really think extending this paper to GNNs (and especially to GGNNs) would add complexity to the theory, confuse the reader and remove the main beauty of the paper, which is dedicated to sequences.
>
> Again, I would like to insist that it is rare to find a combination of theory: an axiomatic derivation of a formula, supported by large scale and relevant experiments.

---

> > ### Comment · AnonReviewer4 · 2019-11-14
> > **Possible Improvements**
> >
> >
> > I agree that the paper addresses an important problem, and proposes a principled approach (solidly motivated through axioms) to the problem.
> > However, the main concern is that the paper seems to be weak w.r.t existing research.
> > As stated in the abstract, the paper addresses the problem of incorporating "the ordered relationship (e.g., adjacency or precedence) between individual word positions" and GNNs seem to be relevant here.
> >
> >
> > Regarding AnonReviewer3's concerns on GNNs:
> > 1) GGNN of [Beck et al., ACL'18] shows empirical improvements in machine translation. Their g2s+ model is precisely the one that uses sequential structure (we can see in Table 2 in their paper that g2s+ outperforms g2s which does not use sequential structure).
> > 2) Extending the theory to GNNs is certainly an interesting future direction!
> > However, a straightforward empirical baseline to compare against seems to be using Transformer - Complex - vanilla embeddings to a GGNN with sequential edges (adjacence, precedence, etc.).
> >
> > [Beck et al., ACL'18] Graph-to-Sequence Learning using Gated Graph Neural Networks
> >
> > As I said before, I am open to revising my rating based on the responses of the authors.

---

> > ### Comment · AnonReviewer1 · 2019-11-14
> > **Great suggestions by R4 but the baselines are sufficient in my opinion**
> >
> > I just wanted to chime in by saying that Reviewer 4 does raise valid concerns about existence of other related methods like GNNs but I feel that the proposed techniques and experiments in the paper are sufficient for the paper to be a standalone piece of work at ICLR. That said, appropriate attribution to GNN based architectures must exist in this paper as form of discussion about related work and future directions. From the author response to R4, I did notice that the authors provide GNN numbers on text classification. I am leaning toward accepting this paper at ICLR.

---

> ### Author Response · Authors · 2019-11-14
> **To Reviewer #4**
>
> Thank you for your comments.
> As suggested by you, we have conducted an experiment on Graph Convolution Networks for text classification (a simple and general NLP task). The results are shown below. The implemented GCN has a two-layer architecture, where each layer encodes and updates all nodes in the graph using features of immediate neighbors based on a dependency tree [1,2]. The experimental results show overall inconsistent, sporadic, and negligible gains, as well as some notable drops in accuracy (trigonometric position embedding setting) on the position embedding setup (including the vanilla learnable position embedding, fixed trigonometric position embedding and the proposed embedding).
>
> |     setting                      |    MR |  SUBJ |    CR |  MPQA |  SST  |  TREC |
> +------------------------------+---------+---------+---------+---------+---------+--------+
> |GCN                                | 0.786 | 0.934 | 0.844 | 0.833 | 0.826 | 0.906 |
> |GCN-PE                          | 0.781 | 0.931 | 0.810 | 0.830 | 0.822 | 0.884 |
> |GCN-TPE                        | 0.548 | 0.928 | 0.656 | 0.828 | 0.818 | 0.886 |
> |GCN-Complex-vanilla | 0.762 | 0.918 | 0.831 | 0.824 | 0.805 | 0.886 |
> |GCN-Complex-order   | 0.781 | 0.931 | 0.825 | 0.833 | 0.816 | 0.900 |
>
> The code used for this experiment is available here:  https://github.com/iclr-complex-order/cgcn .
>
> We agree that GCNs offer a different approach to handle position sensitivity than our proposed method, and we agree that this should be properly reflected  in the paper; we have added a short explanation and  references to Section 3.1.
>
>
> [1] Marcheggiani, Diego, and Ivan Titov. "Encoding Sentences with Graph Convolution Labeling." EMNLP. 2017.
> [2] Zhang, Yuhao, Peng Qi, and Christopher D. Manning. "Graph Convolution over Pruned Dependency Trees Improves Relation Extraction." EMNLP. 2018.

---

> > ### Comment · AnonReviewer4 · 2019-11-14
> > **Thanks for the response**
> >
> >
> > Thanks for the response. I appreciate the additional experiments and the short footnote explanation.  I have increased the rating to weak accept based on the response.
> >
> > What I would like to see is a position-independent word embedding used as initial features to a GCN on the graph with positional edges (words are nodes and the neighbouring words are connected by adjacence, precedence edges). Please note that there are no syntactic edges in such a graph.
> >
> > I think this is a relevant baseline in all the Tables (2, 5.1, and 5.2).  This is the reason for weak accept (and not accept).

---

### Official Review · AnonReviewer3 · 2019-10-30
**Official Blind Review #3**

**Rating:** 8

**Review:**

This paper makes present an original way to encode the position of the token when encoding them in a sequence. The classical additive encoding of positions creates several issues, such as the lack of flexibility when dealing with pooling layers, and the authors refer it as the position-independence problem.

Instead, the proposed approach is based on the encoding of a term-specific frequency (through the complex argument) and modulus in the complex-space, applied once per embedding dimension. This enables the embedding of a word to be dependent on the position in a non-linear manner. The intuition is similar to the use of complex numbers in signal analysis.

Sorry this is not scientifically, but I have to mention that I find the axiomatic derivation of the approach simply beautiful. It is amazing to find such a simple formula from two obvious properties that someone would want from a positional encoding: Position-free offset transformation and boundedness to handle arbitrary length.

The fact that the offset does not have positive effect is interesting, and the discussion about it is limited. I would assume it is due to some redundancy in the other two parameters, but more experiences would be needed.

The rest of the paper shows impressive results, both for text-classification and for machine translation, with a clear comparison with the state-of-the-art. The gains are really significant, providing a clear validation of the approach.

In short, it is quite rare to find such a clear and simple idea with so much empirical gains. I would love to meet the authors once the review period is over.


**Experience Assessment:**

I have published in this field for several years.

**Review Assessment: Checking Correctness Of Derivations And Theory:**

I assessed the sensibility of the derivations and theory.

**Review Assessment: Checking Correctness Of Experiments:**

I assessed the sensibility of the experiments.

**Review Assessment: Thoroughness In Paper Reading:**

I read the paper at least twice and used my best judgement in assessing the paper.

---

> ### Author Response · Authors · 2019-11-14
> **To Reviewer #3**
>
> Thank you for your comments.
> Regarding the fact that the offset does not have a positive effect, we agree with the reviewer and we have added a short discussion about this in the revised paper in Section 3.1. One of the reasons may be that phase is periodical and needs a better way to regularize the initial phase term.

---

### Decision · Program_Chairs · 2019-12-19

**Decision:**

Accept (Spotlight)

**Comment:**

This paper describes a new language model that captures both the position of words, and their order relationships.  This redefines word embeddings (previously thought of as fixed and independent vectors) to be functions of position.  This idea is implemented in several models (CNN, RNN and Transformer NNs) to show improvements on multiple tasks and datasets.

One reviewer asked for additional experiments, which the authors provided, and which still supported their methodology.   In the end, the reviewers agreed this paper should be accepted.